# Sex Differences in the Response to Lung Cancer and Its Relation to Programmed Cell Death Protein-1/Programmed Death-Ligand-1 Checkpoint Therapies

**DOI:** 10.3390/cancers17243953

**Published:** 2025-12-11

**Authors:** Morgan Puglisi, Lauren May, Thusna Gardiyehewa, Joseph W. Landry

**Affiliations:** Department of Human and Molecular Genetics, VCU Institute of Molecular Medicine, Massey Cancer Center, VCU School of Medicine, Richmond, VA 23298, USA; puglisimm@vcu.edu (M.P.); lauren.may1@vcuhealth.org (L.M.); gardiyehewats@vcu.edu (T.G.)

**Keywords:** PD-1, PD-L1, sex-differences, lung cancer, sex hormones, T-cell anergy, NK cell

## Abstract

Immunotherapies have revolutionized lung cancer treatment. The most widely used immunotherapy has been those that target the programmed cell death protein-1/programmed death-ligand-1 (PD-1/PD-L1) checkpoint therapies. These drugs have been FDA-approved since 2015, and as a result, there has been a significant accumulation of data on how patients respond to these drugs. Although not consistently observed, there have been a number of studies that have shown a sex difference in lung cancer incidence and progression, as well as sex differences in response to these drugs. In this review, we summarize evidence in the literature of a possible sex-specific small molecule-driven mechanism underlying sex differences, which could have relevance to the observed sex differences in PD-1/PD-L1 therapy activity in the clinic.

## 1. Introduction

Lung cancer has the highest mortality of all cancers worldwide, with 18.7% of all cancer deaths being due to lung cancer [1]. It remains one of the most commonly diagnosed cancers, second to breast and prostate cancers, with an incidence of 11% for men and 12% for women in 2025 [2]. Between 65 and 84 years old, males have a 4.8% chance of developing lung or bronchus cancer, whereas females have a 4.1% chance of developing lung cancer [2]. Above 85 years old, the difference becomes 2.5% for males and 1.8% for females [2]. Smoking remains the largest risk factor for lung cancer development, contributing to around 85% of lung cancer-related deaths [1,2].

Many studies have attempted to elucidate the reason for the difference in incidence and the course of lung cancer in women versus men. Some studies have shown that female smokers acquired more foreign material in their DNA, which supports data showing female smokers are nearly three times more likely to develop lung cancer compared to male smokers [3]. It has also been speculated that female hormones could play a role in lung cancer development, with some data suggesting exogenous sex steroids may interact with tobacco carcinogens [4]. Many studies have concluded that estrogen can bolster cell proliferation and promote tumor growth, with estrogen state influencing both lung cancer risk and growth of existing malignancies [4]. In addition to autonomous cell effects on lung cancer growth, sex steroids are well known to regulate the immune response to lung cancer, and these activities are believed to play important roles in the observed sex differences in mortality [5].

Mechanisms used by cancer cells to avoid the antitumor immune response include reducing immunogenicity, reducing the activity of immune effector cells, and rendering tumor cells resistant to immune cell killing. Common ways tumors inhibit immune effector cells, including CD8 and CD4 T cells, are in part through inducing immune cell dysfunction. The programmed death-ligand/programmed death-ligand 1 (PD-1/PD-L1) axis has emerged as a major regulator of effector cell dysfunction [6]. The discovery of the PD-1/PD-L1 axis, and how to block it therapeutically, was awarded the Nobel Prize in Medicine in 2018 [6]. Since this original groundbreaking work, PD-1/PD-L1 pathway has been targeted by many FDA-approved therapeutics to control tumor growth of many different malignancies, including breast, lung, liver, and melanoma, to name a few [7]. Pembrolizumab, an anti-PD-1 monoclonal antibody, is now one of the most widely used immunotherapy agents for lung cancer with elevated PD-L1 expression [8]. Other PD-1/PD-L1 agents have been developed, including nivolumab, cemiplimab, toripalimab, atezolizumab, and durvalumab [9]. Pembrolizumab, atezolizumab, nivolumab, and durvalumab have shown improved survival for the treatment of non-small cell lung cancer (NSCLC) when used in conjunction with platinum-based chemotherapy [9].

Shortly after the widespread use of PD-1/PD-L1 therapies began, it became apparent that there was a sex bias in the effectiveness of this therapy [10]. It is our objective in this review to explore the possible mechanisms underlying these observed differences and to propose methods to capture any benefits for the opposite sex.

## 2. Methods

A search of the literature was conducted to complete a narrative review of possible sex-specific mechanisms for PD-1/PD-L1 therapies in lung cancer. Research was conducted for this paper by searching PubMed using the following search terms: sex differences, lung cancer, current PD-1/PD-L1 lung cancer therapies, mechanisms of PD-1/PD-L1 therapies, sex hormone differences in male and female physiology, and sex differences in response to PD-1/PD-L1 therapies in lung cancer. Resources were tracked in Excel and compiled at the conclusion of the drafting process.

## 3. Results

### 3.1. States of T Cell Activity, Its Relation to PD-1/PD-L1, and the Antitumor Immune Response

T cells are the major effector cells of the adaptive immune response. As such, they must be rigorously selected to prevent harm to the host. T cells are selected through both central and peripheral tolerance. Central tolerance occurs in the thymus through both positive and negative selection processes [11]. T cells passing central tolerance that exit the thymus are considered antigen-ignorant and exist in a state of cellular quiescence. While in the periphery, T cells are quiescent until they encounter antigens, which can engage their T cell receptor [12]. This results in T cell priming; however, in many cases, these antigens are self-antigens that activate peripheral tolerance to suppress their activity [12]. Peripheral tolerance occurs to control self-reactive T cells once they pass central tolerance and encounter antigens in peripheral tissues for the first time. Peripheral tolerance can occur when T cell activation is incomplete, meaning it lacks one or more of the signals T cells need for full activation [12]. Conditions that lend to incomplete T cell signaling include the recognition of self-antigens, viral infections, and cancer.

### 3.2. T Cell Activation

T cells require three signals from antigen-presenting cells (APCs) to be fully activated [13]. Signal 1 is primarily an activation signal provided by antigen/MHC class I and II complexes on the APC and the T cell receptor. Signal 1, in the absence of signals 2 and 3, promotes T cell anergy and, ultimately, apoptosis. Signal 2 is a costimulatory signal promoted between CD80 and CD86 on APC and CD28 on the T cell, which alters signal 1 to prevent anergy. Signal 3 is provided by cytokines produced by APCs (a two-cell model) and other immune cells (a three-cell model), which induce T cell differentiation into effector subtypes [14]. All three signals are present on professional APCs, fully activated by ligands for pathogen-associated molecular patterns (PAMPs) or damage-associated molecular patterns (DAMPs). However, under conditions when one or more of these signals are missing, it results in reduced T cell activity or T cell apoptosis. These conditions commonly occur in disease states like chronic viral infections or the tumor microenvironment (TME).

### 3.3. Tumor Microenvironment and T Cell Activity

The tumor microenvironment is populated by T cell states with anti-tumor activity, including effector and stem cell-like states. Their T cell states are inert, including anergy, exhaustion, and senescence (Figure 1) [15].

Effector T cells have been primed for reactivity to tumor antigens in the lymph node and can kill tumor cells presenting a reactive tumor antigen-MHC in the context of costimulation. Productive effector cells can transition to stem-like T cells, which maintain the ability to self-renew and have the potential to generate central memory T cells [15]. Upon restimulation, a central memory T cell can develop into an effector memory CD4+ and CD8+ T cell, which can further become a CD4+ or CD8+ effector T cell [15]. Effector memory T cells and effector T cells can migrate to peripheral tissues to act on infected and malignant cells, producing interleukin-2 (IL-2) and IFN-gamma (IFN-γ) [15]. These antitumor T cell states control tumor growth by directly killing tumor cells. Commonly, tumor cells are edited and selected for their ability to suppress these anti-tumor states, in part through activating intrinsic T cell mechanisms of peripheral tolerance.

Through immune editing, tumor cells acquire the means to suppress effector T cells’ activity by inducing states of anergy, exhaustion, and senescence (Figure 1). Senescent T cells, as the name implies, result from prolonged T cell effector activity and clonal expansion resulting from aging or chronic infection. Senescent T cells are cell cycle-arrested and have shortened telomeres, but they express CD57, KLRG1, and other regulatory receptors [15]. T cell anergy occurs when the environment lacks costimulation (signal 2) or provides inhibition, causing T cells to express checkpoint regulators PD-1 and CTLA4 and enter a state where proliferation and effector ability are not functional [16]. Anergic T cells exhibit decreased production of IL-2 and decreased responsiveness [15]. Different from anergy, chronic stimulation can cause T cells to enter a state of exhaustion, wherein T cells have lost their effector functions and are unresponsive. In this state, inhibitory receptors PD-1, TIM3, and LAG3 are expressed, and co-inhibition is present [16]. While the effector functions of anergic and exhausted T cells are nearly identical, transcription factor expression can differentiate them, with expression of TOX indicating the exhausted T cell phenotype [17].

### 3.4. PD-1/PD-L1 Pathway

PD-1 is a receptor that plays a role in mitigating immune responses and fostering tolerance [18,19]. The receptor is expressed on adaptive immune cells, including T cells, B cells, myeloid dendritic cells (MDCs), monocytes, natural killer (NK) cells, and thymocytes [20]. It is primarily bound by PD-L1, which is expressed by many cell types extra- and intracellularly, including B cells, dendritic cells, macrophages, and T cells, as well as lymphatic and micro-vascular endothelial cells, keratinocytes, and pancreatic islet cells [20,21]. Interactions between PD-1 and PD-L1 play a role in immune tolerance, where they suppress T cell activity, preventing T cells from damaging healthy tissues [18]. Additionally, NK cells can be inhibited by PD-1/PD-L1 pathway. This occurs primarily in an indirect fashion by induction of T regulatory cells (Treg), which subsequently act to inhibit NK cells, but some NK cells in cases of multiple myeloma have been found to express PD-1 as well [22]. PD-L2 can bind PD-1 as well and is expressed primarily by inflammatory macrophages, dendritic cells, bone marrow-derived mast cells, and some B cells. Granulocyte-macrophage colony-stimulating factor (GM-CSF), IL-4, and IFN-γ can also induce PD-L2 expression on macrophages and monocytes [20,23]. Stromal cell fibroblasts can express PD-L1 and PD-L2, often in association with inflammatory states and in tumors [20].

When PD-L1 binds the PD-1 receptor, phosphorylation of the immune receptor tyrosine-based inhibitory motif (ITIM) and immune receptor tyrosine-based switch motif (ITSM), leading to binding of the receptor to phosphatases, Src homology 2 domain (SH2) containing protein tyrosine phosphatases -1 and -2 (SHP-1 and SHP-2) [24]. SHP-1 and SHP-2 decrease antigen receptor signaling through the phosphoinositide 3-kinase (PI3K) pathway by dephosphorylating intermediates, which normally plays a role in propagating cell survival [19,23]. The Akt pathway remains inactivated by PI3K, resulting in a lack of induction of cytokines B-cell lymphoma-extra-large (Bcl-xL), IL-2, IFN-γ, which enhances cell survival [23]. In addition, PD-L1 binding can block Ras activation, signal transducer and activator of transcription 5 (STAT5) phosphorylation, suppressing downstream signals and preventing target gene transcription [25]. The result of this widespread inhibition is decreased immune cell survival and proliferation which leads to T cell apoptosis [25]. This effect can be antagonized by CD80-CD28 costimulation, which fosters immune cell survival [23,26].

The PD-1/PD-L1 pathway is exploited by tumor cells, which utilize PD-L1 to evade the immune response [27]. PD-L1 expression on tumor cells promotes inhibition of downstream pathways necessary for effector cell survival, leading in their inability to target tumor cells [27]. Multiple cancer cell mutations increase PD-L1 expression, as well as expression of IL-6, IFN-γ, and TNF-α in the TME [28]. As a result, this pathway is targeted in the development of new therapeutics to increase cancer susceptibility to the immune system [19].

### 3.5. PD-1/PD-L1 in Lung Cancer

Many studies have been performed to characterize the role of the immune system in lung cancer. Lung tumors have been shown to have very low antigen presentation and low costimulatory molecule expression, allowing them to escape detection by the immune system [29,30]. Patients with NSCLC tend to express higher levels of PD-L1 than other lung cancer subtypes, with increased expression indicating greater susceptibility to anti-PD-L1 treatment options [31]. While increased expression is most prevalent with NSCLC (32.6%), some small cell carcinoma cases (19.6%) demonstrate increased PD-L1 expression [31,32,33]. Increased PD-L1 expression correlates with more aggressive, higher proliferating tumors and those with higher grade and lymph node status [32]. In addition, there is an association with high PD-L1 expression and decreased length of survival in NSCLC, especially in patients with adenocarcinoma [32]. A relationship has been observed between increased PD-L1 and increased p63, which is a transcription factor that functions in epidermal development and typically overexpressed in SCC [32]. Chemotherapy can increase PD-L1 expression, indicating the need for determining the PD-L1 expression at each stage in treatment to determine possible efficacy of anti-PD-L1 therapies [34]. While some argue that increased PD-L1 expression alone indicates greater response to immune checkpoint inhibitor therapy, it is the increased interaction of PD-1 with PD-L1 on tumor cells that correlates with increased susceptibility to immune checkpoint inhibitor therapy in NSCLC [35,36].

### 3.6. T-Cell Exhaustion by PD-L1

Commonly, lung cancer expression of PD-L1 by tumor cells causes chronic T cell stimulation. Eventually, the ability of T cells to recognize and exert an immune response to lung cancer is reduced, which is referred to as T cell exhaustion [25]. A decline in levels of cytokines typically involved in T cell survival and function can also contribute to T cell exhaustion, including IL-2, IFN-γ, and TNF-α [37]. A similar phenomenon is observed in NK cells with high lung tumor expression of PD-L1. NK cells exert an exhausted phenotype in response to constant stimulation, resulting in reduced proliferation, metabolic function, and tumor cell killing [38]. Anti-PD-1/PD-L1 therapy has shown promise in lung cancer for its ability to restore T cell and NK cell function via reversal of the exhausted phenotype [39].

### 3.7. Therapeutic Options

Immune checkpoint inhibitors (ICIs) have been developed to block T and NK cell-inhibitory mechanisms and increase tumor cell susceptibility to the immune response [40]. Anti-PD-1/PD-L1 and anti-cytotoxic T-lymphocyte antigen 4 (CTLA-4) treatments have shown success and have been FDA-approved for several cancer types. Presently, drugs are being developed to target other checkpoint pathways, including the T cell immunoglobulin and mucin-domain containing-3 (TIM-3), lymphocyte activation gene-3 (LAG-3), and OX40 receptors [40].

One of the primary PD-1 inhibitors in use today is pembrolizumab (Table 1), a selective humanized IgG4 kappa anti-PD-1 monoclonal antibody that has shown therapeutic benefit in the treatment of multiple types of cancers [41,42]. It functions as a first-line treatment for PD-L1-positive or metastatic NSCLC, with treatment increasing survival rates and outcomes [42]. KEYNOTE clinical trials have demonstrated efficacy for pembrolizumab as first-line monotherapy in treating NSCLC with increased PD-L1 expression (KEYNOTE-024 and KEYNOTE-042), as well as second-line monotherapy with PD-L1 expression (KEYNOTE-010) [43]. It has also shown efficacy as a first-line combination therapy with platinum doublets (KEYNOTE-189 and KEYNOTE-407) [43]. Current work has focused on the combination of pembrolizumab with other drugs to maximize efficacy in treating NSCLC [41].

Nivolumab is a human IgG4 PD-1 antibody that is effective in treating of variety of cancers, including partial or complete tumor response in NSCLC (Table 1) [44,45]. In the CheckMate 227 Part 1 trial (NCT02477826), nivolumab with ipilimumab (anti-CTLA-4) was found to increase 5-year survival in patients with metastatic NSCLC compared to chemotherapy, although adverse events were increased [46,47].

Cemiplimab is a human IgG4 PD-1 antibody [44] that, when used as monotherapy for advanced NSCLC with ≥50% PD-L1 expression, confers survival benefits and progression-free survival compared to chemotherapy (NCT03088540) (Table 1) [48,49]. The EMPOWER-Lung 3 (NCT03409614) phase 3 trial showed efficacy of dual cemiplimab and platinum-doublet chemotherapy as a first-line treatment for patients with advanced non-small cell lung cancer, revealing increased overall survival, improvement in pain symptoms, and delayed clinically meaningful deterioration [50,51].

Toripalimab is a humanized PD-1 monoclonal antibody used to treat nasopharyngeal carcinoma, advanced NSCLC, and advanced esophageal squamous cell carcinoma (Table 1) [52]. The CHOICE-01 clinical trial (NCT03856411) demonstrated that dual therapy with toripalimab and chemotherapy results in increased progression-free and overall survival in patients with untreated advanced NSCLC. In patients with resectable stage III NSCLC, the Neotorch Randomized Clinical Trial (NCT04158440) demonstrated that toripalimab, when used as an adjunct to perioperative chemotherapy, increased event-free survival [52].

Atezolizumab is a monoclonal anti-PD-L1 antibody that has shown efficacy in treating platinum-resistant metastatic NSCLC and urothelial cancer (Table 1) [53]. The OAK trial (NCT02008227) showed that atezolizumab could improve overall survival in previously treated NSCLC compared to docetaxel [54]. The IMpower010 phase III clinical trial (NCT02486718) showed that atezolizumab after adjuvant chemotherapy in resected stage IB-IIIA NSCLC could improve disease-free survival, especially in patients whose tumors had increased PD-L1 expression [55]. In addition, combination therapy with bevacizumab has shown improved progression-free and overall survival in patients with non-squamous NSCLC, as demonstrated in the Impower150 trial (NCT02366143) [56].

Durvalumab is a human anti-PD-L1 monoclonal antibody that is used to treat extensive-stage small cell lung cancer (ES-SCLC) in combination with etoposide and carboplatin or cisplatin (Table 1) [57]. The phase III CASPIAN trial demonstrated the efficacy of combination therapy with durvalumab in increasing overall survival in patients with ES-SCLC who had not previously been treated [57]. In addition, the phase III PACIFIC trial demonstrated overall and progression-free survival in monotherapy with durvalumab after chemotherapy in unresectable stage III NSCLC [58].

### 3.8. Sex Difference in Treatment Response PD-1/PD-L1 Immunotherapy

PD-1/PD-L1 immunotherapy has quickly become a first-line treatment for multiple cancers. As a result, further studies are unveiling the possible factors that may affect treatment response, including sex. A meta-analysis evaluating sex differences in response to anti-CTLA-4, anti-PD-1, and anti-PD-L1 therapies in combination with immunotherapies in cancer patients showed no significant difference in overall survival (OS) or progression-free survival (PFS) between the sexes. However, one major drawback of this study is the lack of consideration for variability in trials [59]. Similarly, another meta-analysis comparing anti-PD-L1 therapy and chemotherapy found a difference in benefit for males versus females with cemiplimab only; however, the authors report that the limited representation of females may restrict the conclusions made [60]. Wang and colleagues focused on the efficacy of anti-CTLA-4 and anti-PD-1/PD-L1 immunotherapies in NSCLC and found that males had an improved magnitude of clinical benefit from PD-1/PD-L1 therapies but not CLLA4 therapies [61]. A fourth meta-analysis compared anti-PD-1 therapies, pembrolizumab and nivolumab, with chemotherapy in only NSCLC patients. The results demonstrated a clear benefit of anti-PD-1 therapy in males compared to females [62]. In patients with driver-negative advanced or metastatic NSCLC, female patients exhibited a robust objective response rate to first-line combination anti-PD-1 therapy with chemotherapy than men, with greater progression-free survival and overall survival as well [63] (Table 2).

Overall, the sex difference in response to anti-PD-1 therapy has been widely recognized. However, mechanisms for this difference remain under study. Herein, we will propose possible mechanisms by which sex results in variations in response to anti-PD-1 therapy through hormonal differences, immune system variation, and differences in lung cancer characteristics (Table 2).

### 3.9. Sex Differences in PD-L1 Expression in Lung Cancer

There are mixed results on PD-L1 expression in NSCLC in men and women, with some studies indicating a correlation between female sex and increased PD-L1 expression, while others indicate that male sex is associated with increased PD-L1 expression [64,65,66,67,68,69]. In further examination of differences in the tumor microenvironment between sexes, Yu et al. (2024) found that untreated NSCLC tumors in female patients exhibited higher expression of CD4, CD4/FOXP3, and CD4/FOXP3/PD-L1 than tumors from male patients [63]. Gu et al. (2022) found that blood samples from female NSCLC patients revealed decreased serum PD-1 and CD4+ T cell expression of PD-1 compared to female patients without NSCLC [70].

### 3.10. General Differences Between Male and Female Hormone Physiology

In humans, the presence of the Y chromosome leads to the development of male sex via differentiation of the gonad into testes, which eventually produces high levels of anti-Mullerian hormone (AMH) to inhibit the development of female sexual characteristics and increase testosterone to promote the development of male sexual characteristics [71]. In the absence of high levels of AMH or androgens in 46 XX individuals, female sexual characteristics develop [72].

The general differences between males and females are derived from differences in hormone actions and levels between the two sexes [73]. In men, the testes, adrenal glands, and other tissues can produce low levels of estrogen and progesterone, with progesterone serving as a precursor to hormones including testosterone, estradiol, cortisol, and aldosterone [73,74]. In premenopausal women, the ovaries produce estrogen and progesterone, although the adrenal gland and adipose tissue can also produce estrogen [75]. This process is mediated by follicle-stimulating hormone (FSH) and luteinizing hormone (LH), which are released from the pituitary gland under the control of the hypothalamus [75]. In addition, the mid-follicular phase of the menstrual cycle is marked by an increase in estrogen and progesterone to promote the development of the dominant follicle for ovulation in females [76]. Progesterone also contributes to mammary gland development in females and is crucial to many processes in pregnancy [74]. Aromatase is expressed in many tissues, including adipose, muscle, brain, breast, prostate, and liver tissues. It is higher in all tissues in men and functions to convert testosterone to 17-beta-estradiol [77]. Leydig and Sertoli cells can synthesize estrogen, although conversion of testosterone by aromatase in peripheral tissues contributes to most estrogen in circulation [78]. In men, estrogen is thought to play a role in bone health, regulation of spermatogenesis, aggression and sexual behavior, and regulation of insulin signaling [78].

In men, testosterone is primarily produced by the testes with some contribution from the adrenal cortices, and it functions in male reproductive development, development of secondary sexual characteristics, muscle mass, libido, erectile function, cognition, mood, bone health, and erythropoiesis [71,79]. In women, testosterone is produced by the adrenal gland and the ovary, and androgen levels in women tend to exceed estrogen levels, although testosterone can be converted to dihydrotestosterone (DHT) or estradiol [78,80]. Nevertheless, it is thought that in women, testosterone plays a role in bone health, follicle development, sexual desire, and insulin regulation [78]. Gonadotropin-releasing hormone (GnRH) regulation of FSH and LH secretion from the pituitary enables regulation of androgen synthesis, which is stimulated by LH [79].

Prolactin is synthesized by lactotroph cells in the anterior pituitary gland and functions primarily in the regulation of lactation, although it also plays a role in metabolism, bone health, maternal care, adrenal function, and skin and hair follicles in both men and women [81]. The level of prolactin generally remains similar between men and women, although women experience an increased level between ages 20 and 29 [82]. Nursing, light, audition, olfaction, and stress can stimulate prolactin release, whereas dopamine released from the hypothalamus serves as a main inhibitor of prolactin secretion [83].

Activin and inhibin are opposing secreted molecules in the TGF-β superfamily. Activins bind to TGFβ family member cell surface receptors to affect a cellular response. Inhibin does not signal directly, but rather binds to activins, preventing their binding to these receptors. While serum activin is similar between the sexes, inhibin concentrations are significantly different, with males having higher concentrations compared to females [84]. Activin A is a homodimer of inhibin β A (INHBA), which plays a role in inducing FSH secretion, stem cell activity, embryonic development, cell proliferation, fibrosis of tissues, and hematopoiesis [85]. Inhibin A is a glycoprotein composed of inhibin-α and inhibin-β A, whereas inhibin B is composed of inhibin-α and inhibin-β B [86]. In males, inhibin B is produced by Sertoli cells in the testes, with a function in FSH suppression and Sertoli cell function, as well as sperm number and spermatogenesis. In females, inhibin is produced by granulosa cells and the corpus luteum [87]. In both males and females, inhibin is under the control of and regulates FSH. Erythroid differentiation and thymocyte mitotic activity studies have shown that inhibin and activin may also function as growth factors [88]. Boys were found to have a higher baseline level of inhibin B, while girls have a higher baseline of activin A. Sex differences in inhibin B and activin A concentrations are associated with differences in serum FSH levels [89]. The presence of elevated activin A concentrations associated with higher FSH levels in girls may explain the earlier onset of puberty in girls [90]. Although activin and inhibin were originally found to be associated with the gonads, they also play key roles in regulating bone mass and development. Fluctuations in inhibin levels affect osteoblastogenesis and osteoclastogenesis, modulating bone remodeling [91]. Inhibin-α and -β subunits are highly expressed in sex cord-stromal tumors, with some limited expression in mucinous and surface epithelial tumors [92]. Activin A has been found to act as an autocrine growth factor, stimulating the proliferation of sex cord-stromal tumors. It also inhibits progesterone production in these cells, disrupting normal steroid hormone production [93]. When tracing sex differences during embryonic development, activin A was not found to have an impact on developmental competence in the blastocyst stage. However, activin A, as an embryokine, may influence other aspects of embryo function [94].

### 3.11. Role of Sex Hormones in the Immune System

#### 3.11.1. Estrogen Regulation of the Immune System

It is established that immune responses and cell populations are influenced by sex hormones. Estrogen in particular has been known to be widely involved in cellular immunity, including increasing NK cell cytotoxicity in vitro but decreasing NK cell secretion of granzyme B in vivo [95,96]. Estrogen has also been implicated in changes to Th1 and Th2 responses [97]. Additionally, estrogen induces the release of IL-10, which leads to increased levels of human peripheral blood mononuclear cell immunoglobulins [98]. Dunn et al. showed that estrogen is involved in B cell survival and proliferation, as it increases the expression of CD22, SHP-1, and Bcl-2 [99]. It also plays a role in peripheral tolerance as estrogen levels have direct effects on the population size of Tregs, which in turn affects T cell activity [100]. Estrogen also stimulates CD4+ T cell CC-chemokine receptors CCR5 and CCR1 [101]. Differentiation and activation of dendritic cells through myeloid progenitor cell IRF4 production is also impacted by estrogen levels [102]. Non-classical estrogen receptor (ER) signaling has been found to interact with the estrogen response element (ERE) transcription factors NFκB, SP-1, and AP-1 [103,104]. Furthermore, the importance of estrogen in T cell responses was highlighted in a study that found EREs in the promoters of a significant portion of activated genes in female T cells [104]. Another study showed that estrogen and several X-linked microRNAs play a role in modulating PD-L1 expression [105]. It was found that selective ER modulators have the potential to inhibit PD-1 and CTLA-4 [106] (Table 3).

#### 3.11.2. Progesterone Regulation of the Immune System

Progesterone’s effects on the immune system are less convoluted than estrogen’s; progesterone has an overall anti-inflammatory effect, causing decreases in IFN-γ production in NK cells and inducible nitric oxide synthase (iNOS) in macrophages [107,108,109]. Progesterone has also been implicated in the viral response, with induction of antiviral genes and induction of innate antiviral response in cell and mouse studies [110]. In mice studies, it was found that progesterone could bind glucocorticoid receptors and induce CD4+ T cell death, especially at the high levels of progesterone found in pregnancy, suggesting a role in immune tolerance during gestation [111] (Table 3).

#### 3.11.3. Testosterone Regulation of the Immune System

Androgens, such as testosterone, work to suppress immune cell activity and are anti-inflammatory [112,113]. Testosterone has been shown to reduce immunoglobulin production and reduce the production of IL-6 [114]. It also generally suppresses immune responses [115,116,117]. In addition, in vivo and in vitro studies have demonstrated that testosterone treatment induces increases in Treg cell populations. However, testosterone given as gender-affirming therapy for transgender men results in increased TNF, IL-6, and IL-15 production and activation of NF-κB genes by influencing monocyte activity [118].

During the ovulatory phase in women, there is an androgen-dependent increase in Foxp3 expression, suggesting a role of androgens in Treg cell differentiation [119]. Studies have shown that low androgen levels are associated with a decreased ability to exert an inflammatory response, and this is thought to be associated with the fact that the presence of androgens increases Foxp3 expression [120]. In addition, in transgender men, testosterone injection continuously decreases the production of interferon type 1 (IFN-1) by plasmacytoid dendritic cells (pDCs) [121]. Comparative studies with hypogonadal men show testosterone levels are inversely correlated with antigen-stimulation CD107b expression on CD16+ cells in men, suggesting men with hypogonadism have heightened immunologic responses and risk for autoimmune conditions [122] (Table 3).

#### 3.11.4. Follicle-Stimulating Hormone Regulation of the Immune System

FSH has been shown to stimulate TNF production in bone marrow granulocytes and macrophages [123]. In addition, FSH receptors are expressed on monocytes, indicating that they may play a role in the inflammatory response, although isolated CD4+ T cell culture studies have suggested that FSH stimulation does not promote inflammation [124]. Experiments in bone marrow cultures revealed that FSH presence led to an increased proportion of monocytoid CD14+/RANK+ and B cell CD19+/RANK+ cells [125]. Others have shown a negative impact of FSH and its receptor on decidual mesenchymal stem cells, acting through MyD88 pathways to decrease IL-6, which is thought to offer an advantage to immune tolerance in pregnancy [126] (Table 3).

#### 3.11.5. Leptin Regulation of the Immune System

Leptin, which is generally higher in females, can affect the survival and activation of B and T cells, and it can increase IL-2 and IFN-γ production [127]. Leptin is generally increased in the setting of infection and inflammation, and the leptin receptor (Ob-R) has shown the ability to activate the JAK-STAT, PI3K, and MAPK signaling pathways by acting similarly to IL-6 receptors [128]. In vitro studies have revealed that human leptin can work to increase monocyte proliferation and activation by increasing surface markers, with further studies suggesting that leptin may act as a monocyte growth factor. On dendritic cells, leptin has been found to increase IL-8, IL-12, IL-6, and TNF-α production and possibly contributes to dendritic cell survival and maturation [129]. Leptin also demonstrates the ability to promote the survival of human polymorphonuclear neutrophils (PMNs) and plays a role in chemotaxis [128]. In NK cells, leptin can activate STAT3 phosphorylation and increase gene expression of IL-2 and perforin, serving as a regulator of NK cell function [130]. Furthermore, studies on ob/ob mice have revealed that leptin has a role in maintaining thymic structure and function as well as promoting T cell survival [128] (Table 3).

#### 3.11.6. Prolactin Regulation of the Immune System

Prolactin also has effects on the immune system; it has been shown to promote CD4 and CD8 T cell differentiation, the release of IFN-γ from NK cells, and macrophage activation via prolactin receptor activity [131]. In addition, it can alter the production of cytokines of Th1 and Th2 type and promote IL-6 secretion as well as moderate levels of IL-2 [132]. Prolactin has also been implicated in autoimmune conditions due to its inhibition of negative selection of autoreactive B lymphocytes [132] (Table 3).

#### 3.11.7. Luteinizing Hormone Regulation of the Immune System

LH has been implicated in the viral response along with GnRH, increasing in viral infections to cause subsequent increases in progesterone [110]. It also plays a role in hematopoietic stem cell homeostasis during puberty, with low levels of LH receptor serving as a key factor in limiting expansion [133]. It is also thought that LH contributes to the regulation of the immune response in pregnancy, permitting immune tolerance of the fetus [134] (Table 3).

#### 3.11.8. Activin A and Inhibin Regulation of the Immune System

Activin A is found to be elevated in inflammatory bowel disease and inflammatory arthropathies. In vitro studies showed that activin A can have both pro- and anti-inflammatory effects by mediating key cytokines, including TNF-α, IL-1β, and IL-6 [135]. Activin was found to directly increase monocyte chemotaxis and activity. Inhibin and activin compete for effects on IFN-γ production, with inhibin decreasing production and activin increasing production [136]. Additionally, activin A and inhibin A were found to be crucial in pregnancy to prevent rejection of the embryo by interfering with dendritic cell maturation to promote tolerance and prevent activation of allogeneic T cells by the embryo [137]. Given that activin A plays a role in spermatogenesis and testicular steroidogenesis regulation, combining this with its broad implications in inflammation and immunity control, activin A may also play a part in providing testicular immune privilege [138]. Activin A influences the immunological microenvironment of the testes by controlling the activity and phenotype of resident macrophages [139]. In the context of malignancies, longer overall survival was seen in gastric and esophageal adenocarcinoma patients with higher expression of activin subunit inhibin beta A (INHBA). In these patients specifically, high CD4+ T cell infiltration was also observed, suggesting a correlation between activin and tumor immune response. However, in melanoma patients, activin A was found to be inhibitory to the antitumor immune response by affecting CD8+ T cell infiltration indirectly. It was also found that INHBA expression was associated with resistance to anti-PD1 therapy [140]. Taken together, this reflects the highly variable effects on immunity that activin and inhibin can have depending on the cancer in question (Table 3).

### 3.12. Hormones in the Immune Response to Lung Cancer

#### 3.12.1. Estrogen and Estradiol in the Response to Lung Cancer

In vitro studies of NSCLC suggest that tumors can produce estradiol using aromatase, and in ER-positive NSCLC, estradiol can play a role in tumor growth through this mechanism [141]. In vivo studies with aromatase inhibitors and cisplatin therapy in treating NSCLC show a significant reduction in tumor progression with monotherapy or combination therapy with aromatase inhibitors [142]. Phase II clinical trials with aromatase inhibition letrozole have attempted to determine its ability to increase susceptibility to treatment, but incidents of pulmonary toxicity have limited trials [143]. Biomarker studies from a Phase 1b trial with aromatase inhibitor exemestane showed that the overall response rate was correlated with tumor aromatase expression in postmenopausal women with stage IV non-squamous NSCLC, and the use of exemestane was well-tolerated [144]. These results are supported by in vitro studies of normal lung fibroblast and cultured NSCLC cells, which revealed that estradiol can induce the growth of tumor cells [145]. It is conceivable that lung cancers could respond directly to estrogens because several reports have shown that lung cancers express estrogen receptors [145,146]. In addition, lung fibroblast secretion of hepatocyte growth factor could be elevated via treatment with estradiol [145].

Cell studies of lung adenocarcinoma have revealed that estrogen can induce overexpression of CXCR4, which could support tumor growth and metastasis through the CXCL12/CXCR4 pathway (Figure 2A) [147]. In vivo studies of mice with NSCLC showed that ERβ can increase NSCLC cell invasion, supporting a linkage between ERβ/circ-TMX4/miR-622/CXCR4 signaling and NSCLC invasion [148]. Studies of NSCLC have revealed that increased expression of estrogen receptor 1 (ESR1) is correlated with decreased CD4+ and CD8+ activated T cell infiltration of tumors, and increased immune checkpoint markers were found in patients who had increased ESR1 [149]. In addition, higher ERα expression can contribute to increased activity of infiltrative macrophages via activation of the CCL2/CCR2 pathway [150]. This promotes a positive feedback loop in which lung cancer cells upregulate the CXCL12/CXCR4 pathway to further increase ERα expression [150]. Sex differences in ERβ, CXCL12, and CXCR4 expression have also been demonstrated in lung adenocarcinomas, with tumors from premenopausal women having greater expression than tumors from postmenopausal women and men (Figure 2A) [151].

Furthermore, it has been shown in premenopausal women with lung cancer that estradiol can have inhibitory effects on p53 via inhibition of DNMT1, leading to M2-macrophage polarization and poorer prognosis [152]. The use of high-dose therapy of ER blocker tamoxifen was found to increase one-year and median survival when used to treat stage III-IV NSCLC in combination with chemotherapy compared to patients who only received chemotherapy [153]. These results are supported by another randomized control trial on advanced NSCLC already treated with platinum-based first-line chemotherapy, which found that adding tamoxifen to docetaxel therapy resulted in a better response rate and disease control rate than with chemotherapy alone [154].

In addition, the use of anti-estrogen drugs in patients with NSCLC has been shown to decrease the size of tumors and decrease cell proliferation [155]. In a study of male mice, estrogen administration could inhibit tumor growth, decrease NF-κB immunosuppression, and increase immune system response against the tumor in the Kras mutant situation [156]. In addition, bilateral oophorectomies of female mice led to increased tumor burden and NF-κB response, suppressing the immune system reaction against the tumor, suggesting estrogen’s antitumor role can be exercised through NF-κB inhibition [156]. However, a cohort study of women who had premenopausal bilateral oophorectomy found that it did not decrease the risk of de novo lung cancer [157]. In contrast to the above, RNA-sequencing and tumor microenvironment analysis studies have found that inhibiting ER signaling promotes a pro-tumor microenvironment in Kras mutant tumors, with evidence for estrogen in STAT3 signaling [158].

One study found that ERα upregulated PD-L1 throughout transcription, likely contributing to the fact that the use of aromatase inhibitors was found to increase the efficacy of pembrolizumab [157]. It has even been suggested that there is a loop in which tumor cell production of 17β-estradiol via aromatase activates ERα, and NSCLC patient response to pembrolizumab can be predicted by determining 17β-estradiol/ERα status [159]. In vivo studies reveal that estradiol elevated PD-L1 and promotes immune evasion in NSCLC by upregulating the ERβ/SIRT1 axis, leading to increased PD-L1 expression and increased growth and metastasis of NSCLC with administration of estradiol [160] (Table 3).

#### 3.12.2. Progesterone in the Response to Lung Cancer

In vivo studies revealed that progesterone can inhibit cell growth in lung adenocarcinoma via membrane progesterone receptor alpha (mPRα), with higher levels of mPRα expression correlating with worse prognosis and downregulation of mPRα, leading to inhibition of lung adenocarcinoma proliferation [161]. Studies of patients with NSCLC revealed increased progesterone receptor (PR) to be associated with lower attack of tumor cells by activated CD4+ and CD8+ T cells [149]. In addition, analysis of tumor tissue samples from NSCLC stage I-IIIA revealed that in both male and female patients, increased PR expression in stromal cells surrounding tumors was correlated with improved disease-specific survival [162]. In female patients, tumor expression of PR can be used as a prognostic indicator of disease-specific survival [162] (Table 3).

#### 3.12.3. Testosterone in the Response to Lung Cancer

Increased testosterone correlates with decreased membrane-bound PD-1 expression on T cells, which was mediated through increased testosterone levels in female patients with NSCLC (Figure 2A) [70]. A meta-analysis of androgen deprivation therapy (ADT) in males with NSCLC shows that ADT significantly improved OS, suggesting a positive role of male androgens in NSCLC progression [163] (Table 3).

#### 3.12.4. Follicle-Stimulating Hormone and Luteinizing Hormone in the Response to Lung Cancer

Stimulation of human lung cancer cells with FSH led to downregulation of heme oxygenase-1 (HO-1) expression, a survival molecule for cancer cells, and increased the ability of the lung cancer cells to infiltrate other sites in the body, suggesting a role in metastasis [164]. Like FSH, stimulation of human lung cancer cells with LH led to downregulation of HO-1 expression and increased the ability of the lung cancer cells to infiltrate other sites in the body (Figure 2B) [164] (Table 3).

#### 3.12.5. Prolactin in the Response to Lung Cancer

Prolactin in lung cancer takes on a different role from other hormones, as lung cancers can ectopically secrete prolactin, increase general levels, and correlate with more aggressive cancers [165]. In fact, an observational study found that hyperprolactinemia could represent an early predictive factor for decreased response of NSCLC to nivolumab [166].

However, like FSH and LH, stimulation of human lung cancer cells with prolactin led to downregulation of HO-1 expression and increased the ability of the lung cancer cells to infiltrate other sites in the body (Figure 2B) [164] (Table 3).

#### 3.12.6. Leptin in the Response to Lung Cancer

Increased leptin levels have been correlated with reduced risk of death in patients with advanced NSCLC [167]. Cancer-associated fibroblasts express more leptin than normal lung fibroblasts, with high leptin and Ob-R expression in NSCLC cells as well [168]. In addition, it is suggested that leptin produced by cancer-associated fibroblasts activates MAPK/ERK1/2 and PI3K/AKT pathways in NSCLC, promoting proliferation and migration of malignant cells [168]. Patients with adenocarcinoma of the lung were found to have higher serum leptin levels compared to patients with squamous cell carcinoma and other types of NSCLC. However, meta-analyses have failed to detect associations between serum leptin levels and NSCLC [169].

Interestingly, mice and clinical experiments have revealed a positive correlation between obesity and anti-PD-1/PD-L1 therapy in cancer patients, leading to greater survival after therapy, despite higher rates of immune dysfunction and tumor progression [170] (Table 3).

#### 3.12.7. Activin A and Inhibin in the Response to Lung Cancer

Activin A is elevated in patients with lung cancer [171] and is associated with reduced survival [172]. Lung cancers can express activin, which contributes to sarcopenia [173]. In this same report, the authors showed that there was a trend for lung cancers in females, inducing increased levels of activin secretion compared to males. Activin A secretion is induced in alveolar macrophages in lung cell carcinoma due to increased INHBA expression, and the use of activin A antagonists is found to inhibit the proliferation of malignant cells, suggesting that activin A supports lung cancer cell development [174]. In mice, it was found that increased expression of tumor INHBA led to decreased PD-L1 expression and decreased efficacy of anti-PD-L1 therapy, atezolizumab, which is thought to work via decreased signaling through IFN-γ [175]. These studies show that activin A and INHBA are promising therapeutic targets for enhancing response to PD-1/PD-L1 immunotherapy [175]. However, it was also found that activin A can enhance anti-tumor immunity in the lung by limiting T cell exhaustion, re-enabling them with effector properties (Figure 2C) [176].

High expression of INHBA in NSCLC was found to be associated with poor differentiation and advanced tumor stage, as well as decreased five-year overall survival. In vitro studies showed that INHBA overexpression inhibited the Hippo pathway, promoting invasion in NSCLC. This shows that INHBA expression in NSCLC may facilitate tumor development and can determine prognostic outcomes [177]. In NSCLC, it was found that NF-κB was necessary for upregulating activin to induce epithelial-to-mesenchymal transition by acting as an autocrine factor, in turn promoting metastasis [178]. In lung adenocarcinoma, it was found that there is lower baseline expression of activin A, contributing to its occurrence, progression, and malignancy [179]. Interestingly, treating lung adenocarcinoma with TGFβ1 suppressed tumor growth and transformation of cell phenotypes by maintaining intracellular actin filament organization; however, treating with activin A did not have the same effect. In primary lung adenocarcinoma, it was found that expression levels of INHBA were 3 times higher than in normal lung tissue. Upon treatment with small-interfering RNA targeting INHBA, cancer cell proliferation decreased [180] and improved the therapeutic effects of radiation therapy [181]. Furthermore, it was found that the growth of alveolar macrophages is associated with the proliferation of lung cancer cells, and under tumor-bearing conditions, they express high levels of INHBA. INHBA expression leads to an increase in activin A secretion. Subsequently, the use of an activin A inhibitor, FSH, was able to suppress cell proliferation [174]. Activin A was found to inhibit human lung adenocarcinoma cell proliferation, induce apoptosis, and upregulate markers of ER stress (CHOP), DNA damage (GADD34), and apoptosis (cleaved-caspase-3) (Figure 2C) [182] (Table 3).

**Table 3 cancers-17-03953-t003:** Summary of sex-specific molecules and their effects on the immune system and lung cancer.

Molecule	Abundance	Effects on Immune System	Effects on Lung Cancer
Estrogen	Highest in females, with increased abundance during the follicular and leuteal phases, and decreased abundance at menstruation. Low after menopause.	• Modulates NK cell cytotoxicity [95,96] • Affects Th1 and Th2 responses [97] • Increases production of IL-10 [98] • Promotes B cell survival [99] • Affects Tregs [100] • Can stimulate CD4+ T cell CCR5/CCR1 [101] • Affects dendritic cell differentiation and activation via IRF4 [102] • Modulates estrogen response elements in T cells [104] • Can modulate PD-L1 expression through mRNAs [106]	• Produced by lung cancers to drive growth [141,145] • Lung cancers express estrogen receptors [145,146] • Induces overexpression of CXCL12/CXCR4 pathway to support tumor growth and metastasis [147,148,150,151] • Inhibits p53 through DNMT1 [152] • Has antitumor roles via regulating NFkB [156] • Increases PD-L1 expression [157,160]
Progesterone	Highest in females, with increased abundance during the leuteal phase, and decreased abundance at menstruation. Absent after menopause.	• Is anti-inflammatory to NK cells and dendritic cells [107,108,109] • Can bind glucocorticoid receptors to induce CD4 T cell death [111]	• Can inhibit lung cancer proliferation [161] • Tumor cell PR reduces T cell toxicity [149]
Testosterone	Highest in males, with decreasing abundance after 50 years old.	• Has anti-inflammatory roles [112,113] • Can reduce IL-6 and immunoglobulin secretion [114] • Enhances overall suppression of immune responses [115,116,117] • Can regulate Treg cell differentiation [119,120]	• Increases T cell PD-1 expression [70]
Follicular-Stimulating Hormone (FSH) and Luteinizing Hormone (LH)	Highest in females, with increased abundance during ovulation. High after menopause.	• FSH stimulates TNF production from bone marrow granulocytes and macrophages [123] • Increases monocytoid and B cells [125] • Deceases IL6 secretion mesenchymal cells [126]	• Downregulates anti-apoptotic HO-1 [164]
Leptin	Generally higher in females.	• Affects survival and activation of B and T cells and drives IL-2 and IFNγ production [127] • Increases IL8, IL12, and TNFα production in dendritic cells [129] • Increases survival of polymorphonuclear neutrophils and T cells [128] • Regulates NK cell activity via IL2 production [130]	• High expression reduces death from lung cancer [167] • Stimulates ERK and AKT pathways, promoting proliferation and migration [168]
Prolactin	Generally higher in females.	• Promotes T cell differentiation and promotes release of IFNγ by NK cells [131] • Alters Th1 and Th2 cytokine production [132] • Inhibits autoreactive B cells [134]	• Secreted by lung cancers to promote growth [165,166] • Downregulates anti-apoptotic HO-1 [164]
Activin/Inhibin	Activin—similar between females and males. Inhibin—highest in males and in females in the follicular phase. Low after menopause.	• Has both pro- and anti-inflammatory effects via cytokine regulation [135] • Increases production of IFNγ [136] • Promotes tolerance to fetus [137] • Promotes immune privilege in the testis [138,139]	• Elevated in lung cancer patients, associated with reduced survival [171,172] • Can cause sarcopenia [173] • Increases INHBA expression, decreases PD-L1, and reduces anti-PD-L1 therapy [175] • INHBA inhibits Hipo to increase invasion [177] • INHBA is elevated in lung cancers and, when inhibited, decreases proliferation and improves therapy [180,181] • Can inhibit lung cancer proliferation and increase apoptosis [182]

### 3.13. Role of Sex Hormones in the Response to PD1/PDL1 Therapies

Recent studies showed ERβ is the dominant ER form in PD-L1-NSCLC tumors, whereas ERα expression is elevated in PD-L1+ NSCLC tumors. Each of these observations was independent of sex or hormone status (menopause); however, ERα expression was highest in premenopausal women. The lack of correlation with sex could be because many lung cancers express estrogens endogenously through aromatase expression [183]. ERα expression was not associated with PD-L1 tumor proportion score or clinical features. In contrast to ER status, there was low AR expression in each group, and there was no correlation with PD-L1 status. These results suggest that ERα, but not AR, could play a role in PD-L1 expression in clinical samples [184]. These results are supported clinically, where ER status is predictive of the response to pembrolizumab independent of gender and PD-L1 levels [159].

Beyond estrogens and androgens, there is limited data on leptins and the activin-binding protein follistatin in the efficacy of PD-1/PD-L1 in lung cancer therapy. Serum levels of follistatin were significantly lower in NSCLC patients with a durable clinical benefit to PD-L1 therapy compared to those without a durable benefit. Also, serum leptin levels were significantly lower in patients who had adverse immune-related events compared to those without adverse immune events [185]. There is no data available on progesterone, activins, inhibins, FSH, and LH, suggesting the need to conduct targeted studies.

## 4. Discussion

Evidence shows there is a sex difference in lung cancer, with females having reduced cancer progression and mortality compared to males [2,5]. The mechanisms behind this effect are likely complex, but evidence points to biological differences between the sexes being the major cause [186]. Several studies have demonstrated statistically significant differences between sexes in response to PD-1/PD-L1 therapies, but the effects have been mixed overall. Most studies show that males have increased PD-1/PD-L1 expression on tumors with increased susceptibility to such agents, while others suggest women with advanced disease are more positively affected by these therapies when used in combination with chemotherapy [5]. Given the significant differences in response between males and females, it is reasonable that sex hormones could be contributing to this variation. The mechanism by which these biological differences contribute to lung cancer differences and sensitivity to PD-1/PD-L1 therapies is likely multi-factorial, with cell autonomous and non-cell autonomous immune effects contributing. Ideally, understanding the mechanisms at play for sex differences in lung cancer growth, and with immunotherapies, can help to improve overall lung cancer prognosis for males and make immunotherapies more effective for females.

A first step toward this goal would be to identify bioactive molecules that confer the observed differences through cell-autonomous effects. Possible scenarios include an anti-tumor factor present in females that directly inhibits tumor growth. Conversely, there could be a pro-tumor factor present in males that directly enhances tumor growth. Unfortunately, the evidence suggests that there is no single sex-specific molecule that would have direct tumor effects to explain the sex difference. The most obvious candidate would be estrogen, with some data suggesting that the NSCLC response to pembrolizumab can be predicted by the activity of the 17-β-estradiol/ERα loop [159]. Lung cancer cells express estrogen receptors, and much of the evidence suggests that estrogens have cell-autonomous pro-tumor functions, not anti-tumor, in females. There is some evidence that progesterone could have cell-autonomous, anti-tumor functions in females, but the differences in serum progesterone concentration in the serum between males and females are different between the sexes only during ovulation, suggesting that progesterone’s effects would apply unequally over a reproductive life span [187]. FSH, LH, and prolactin can all downregulate the anti-apoptotic factor HO-1 in lung cancer cells, possibly resulting in reduced tumor growth [164]. However, the concentration of each molecule is similar between females and males before menopause, with transient differences occurring during menstruation, pregnancy, and lactation [188]. However, post-menopausal women experience a dramatic increase in serum FSH and LH levels compared to men, which could be relevant to lung cancers, which tend to occur after 50 years of age [187]. The evidence of sex-specific molecules conferring a sex difference in the cell-autonomous survival of lung cancer is complex. This lack of evidence points to non-cell autonomous mechanisms having a more significant role.

A lack of clear sex differences in cell-autonomous mechanisms suggests that the non-cell-autonomous immune response could confer an increased survival benefit to females over males. Experimental data has shown significant differences in the tumor microenvironment of NSCLC in men versus women, suggesting a difference between sexes driving discrepancy in immune response [189]. Sex differences in the expression of immunogenetic factors in NSCLC could lead to differences in immune surveillance and ICI response [190]. Sex differences in tumor cell antigenicity could also enhance T cell-mediated anti-tumor immunity. In some cases, tumors (liver, kidney, melanoma, bladder) have greater mutation burden in males compared to females; however, this is not the case for lung cancer [191,192]. However, tumor mutational burden may be a mitigating factor in the response of NSCLC to ICIs. Newer data suggests that the sex difference in response to ICIs is observed in tumors with low mutation burden, and that no sex difference is observed with tumors of higher tumor burden [193]. It is clear that several components of the immune system are more efficient in females compared to males, including adaptive (T and B cell) and innate (NK cell, neutrophil, and macrophage) cells [194]. These enhancements could be because estrogens enhance immune cell activity, and testosterone inhibits immune cell activity. However, it is thought that some aspects of the female immune system are geared toward optimizing immune tolerance during gestation, which could lead to decreased aggressiveness of immune responses to lung cancers [111]. There is a growing body of evidence that androgens induce T cell exhaustion, which would serve to inhibit antitumor immune responses to many cancers, including lung cancer [195,196,197]. In addition, sex differences could be due to differences in the sensitivity to immune cell killing. We recently showed that lung cancer cells respond differently to female serum compared to male serum, making them more sensitive to TRAIL-mediated immune cell killing [198]. It is plausible to predict that many of these effects, in combination, could confer the observed differences in lung cancer progression.

Other sex-regulated molecules could also be involved. Less is known about the roles of other sex-specific molecules, like FSH, LH, and activin, in PD-1/PD-L1 and other checkpoint therapies. Activins regulate immune cell activity and the response to PD-1/PD-L1 therapies [175]. While serum activin levels do not vary greatly between the sexes, there is a large sex difference in inhibin levels, which would restrict activin activity for females [89]. In addition, lung cancers can express activin, and with males having increased inhibin in the serum, that activin would be more functional in females [173]. Activins/inhibins’ role in lung cancer needs further study to determine their importance for sex differences in tumor growth and immunotherapy responsiveness.

This review on the role of sex hormones in the efficacy of PD-1/PD-L1 remains limited by the current experimental data. Given the nature of these trials, it can be challenging to have a large sample size with an equal split between sexes. Many studies are limited by small sample sizes and declining numbers of women, leading to less reliable results and making it less likely to obtain a statistically significant result. As a result, some studies may have found a difference in treatment between sexes, more so due to a difference in sample size and other confounding variables, whereas other studies may not have been able to confidently support a difference due to decreased sample sizes. There is even less data on the relationships between sex hormones and PD-1/PD-L1 expression in particular. Further research on these topics could elucidate other avenues by which tumors could be made susceptible to immunotherapies.

While many reviews have acknowledged the differences in sex in response to PD-1/PD-L1 immunotherapy, few studies have attempted to elucidate the underlying hormonal mechanisms by which these differences occur. In this review, we assembled relevant literature to identify possible influences of sex bioactive molecules on clinically relevant ICIs to elucidate a mechanism for sex differences in response to these agents. Given recent data of an initial cohort, a multimodal approach (genomics, radiology, and pathology) better predicts response to ICIs [199]. A similar multimodal approach, which includes serum hormone/bioactive molecule levels and is powered to detect sex differences, could be necessary to identify relevant bioactive molecules. By determining this mechanism, steps can be taken toward addressing these differences to enhance the efficacy of ICI therapies for the different sexes.

## 5. Conclusions

In this review, we summarized the roles of sex-specific bioactive molecules (estrogens, progesterone, testosterone, FSH, LH, prolactin, leptin, activin, and inhibin) in the immune response to lung cancer and immunotherapy effectiveness. Many of these sex-specific bioactive molecules modulate the immune system and the immune response to lung cancer. There are well-documented roles for estrogens and testosterone, modulating the immune response and PD-1/PD-L1 to lung cancer. However, evidence has also emerged for the roles of other sex-regulated molecules in these effects, including FSH, LH, activins, and inhibin. Further research needs to be performed to elucidate the relative importance of these molecules in contributing to treatment outcomes between men and women with lung cancer treated with PD-1/PD-L1 immunotherapies.

## Figures and Tables

**Figure 1 cancers-17-03953-f001:**
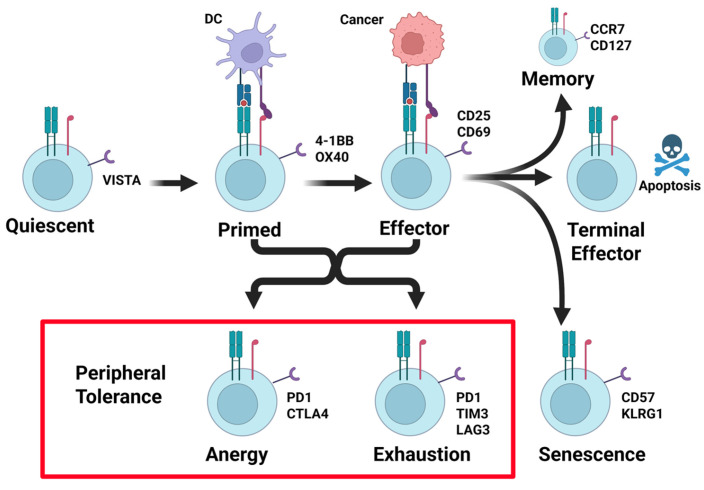
Stages of T cell activation and peripheral tolerance suppression. T cells leave the thymus after passing central tolerance in a quiescent antigen-naïve state. T cells are primed after they encounter antigen–MHC complexes in the context of costimulatory and cytokine signals. Re-encounter with antigen–MHC complexes in the context of costimulatory signals results in effector cell activity. Once resolved, terminal effector differentiation results in apoptosis of the majority of effector cells, with the exception of a small number of long-lived memory T cells. Cancers frequently induce mechanisms of peripheral tolerance. This occurs from antigen stimulation in the absence of costimulation, or chronic antigen + costimulation signals, resulting in T cell anergy or exhaustion (see red box). Each of these states is identified in part by the upregulation of the T cell-suppressive receptor PD-1 (see red box). Created in BioRender. Landry, J. (2025) https://BioRender.com/6kfzt3t (accessed on 15 March 2025).

**Figure 2 cancers-17-03953-f002:**
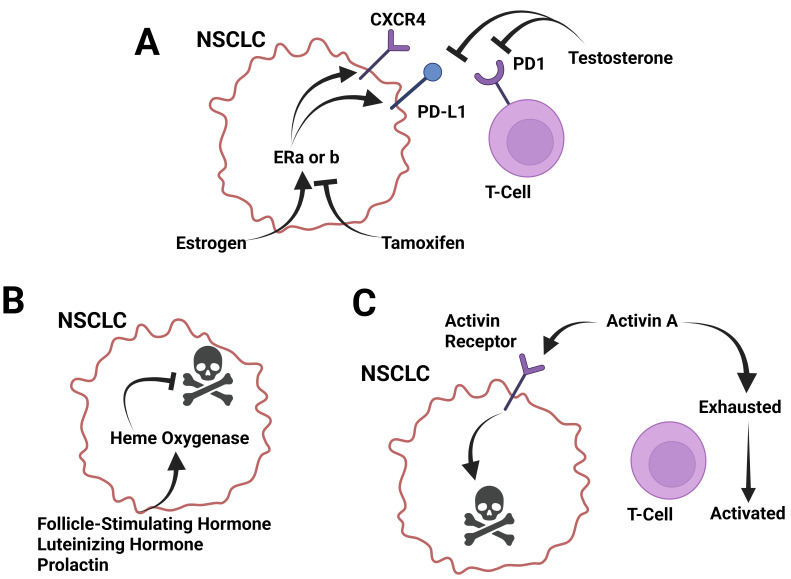
Known roles of sex-regulated molecules in lung cancer progression. (**A**) Sex hormones estrogen and testosterone have counter roles in regulating lung cancer growth and PD-L1 expression. Estrogen induces CXCR4 and PD-L1 expression. The increase in CXCR4 promotes enhanced immune cell-mediated killing. These effects are countered by the estrogen receptor antagonist tamoxifen. Testosterone is a suppressor of PD-1 and PD-L1 and but acts as a pro-growth signal. (**B**) Follicle-stimulating hormone, luteinizing hormone, and prolactin all induce heme oxygenase expression, which is an anti-apoptotic factor. These factors act as pro-growth signals for lung cancer. (**C**) Activins induce apoptosis in lung cancer cells through activin receptor signaling and inhibit T cell exhaustion. These effects, in combination, inhibit lung cancer growth. Created in BioRender. Landry, J. (2025) https://BioRender.com/se7afyn (accessed on 15 March 2025).

**Table 1 cancers-17-03953-t001:** List of FDA-approved PD-1/PD-L1 immunotherapies. Included is a summary of major clinical trials using FDA-approved PD-1/PD-L1 immunotherapies to treat lung cancers.

Drug	Antibody Type	Lung Cancer Type	Current Stage	Clinical Trials
Pembrolizumab	Humanized IgG4 PD-1 antibody	PD-L1+ or metastatic NSCLC	Phase IV	KEYNOTE 010, 024, 042, 189, 407
Nivolumab	Humanized IgG4 PD-1 antibody	NSCLC	Phase IV	Checkmate 227
Cemiplimab	Humanized IgG4 PD-1 antibody	NSCLC	Phase IV	EMPOWER-Lung1
Toripalimab	Humanized PD-1 antibody	Advanced NSCLC	Phase III	CHOICE-01 CHOICE 03 Neotorch
Atezolizumab	Humanized IgG1 PD-L1 antibody	Platinum-resistant NSCLC	Phase IV	OAK IMpower010
Durvalumab	Humanized IgG1 PD-L1 antibody	ES-SCLC NSCLC	Phase IV	CASPIAN PACIFIC

**Table 2 cancers-17-03953-t002:** List of meta-analyses of clinical trials using immunotherapies on lung cancer patients. Summary of major meta-analyses of clinical trials using immunotherapies to treat lung cancers is included.

Meta Analysis	# Clinical Trials	Therapies	# Patients	Sex Difference Lung Cancer
Lee et al. [59]	CheckMate 017, CheckMate 057, KEYNOTE 010 OAK POPLAR	nivolumab, pembrolizumab, atezolizumab	3025	No
Wu et al. [60]	CheckMate 017, CheckMate 026, CheckMate 057, TASUKI-52, KEYNOTE 024, KEYNOTE 042, KEYNOTE 189, KEYNOTE 407, EMPOWER-Lung1, EMPOWER-Lung3, CameL-sq, CameL, RATIONALE 303, RATIONALE 304, RATIONALE 307, ORIENT-3, ORIENT-11, ORIENT-12, CHOICE-01, ASTRUM-005, IMpower110, IMpower130, IMpower 131, IMpower 132, IMpower 150, OAK JAVELIN Lung 200, PACIFIC MYSTIC POSEIDON CAPSTONE-1, GEMSTONE-301, GEMSTONE-302	nivolumab, pembrolizumab, cemiplimab, tislelizumab, sintilimab, toripalimab, serplulimab, atezolizumab, avelumab, durvalumab, adebrelimab, sugemalimab	11,883	Yes, cemiplimab only
Wang et al. [61]	CheckMate 227, IMpower 131, IMpower 132, JAVELIN Lung 200, KEYNOTE 042, KEYNOTE 189, KEYNOTE 407, PACAFIC CA184-104, CheckMate 026, OAK KEYNOTE 010, KEYNOTE 024, CheckMate 057, CheckMate 017	nivolumab, atezolizumab, avelumab, pembrolizumab, durvalumab, ipilimumab	9583	Yes
Conforti et al. [62]	KEYNOTE-042, KEYNOTE-024, IMpower110, EMPOWER-Lung 1	pembrolizumab, atezolizumab, cemiplimab	1672	Yes

## Data Availability

No new data were created or analyzed in this study. Data sharing is not applicable to this article.

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
