# Peer review of "Sex Differences in the Response to Lung Cancer and Its Relation to Programmed Cell Death Protein-1/Programmed Death-Ligand-1 Checkpoint Therapies"

_cancers, 2025, doi:10.3390/cancers17243953_

Round 1

Reviewer 1 Report

Comments and Suggestions for Authors

In this review, the authors wrote comprehensive overview of sex-related differences in lung cancer and PD-1/PD-L1 immune checkpoint treatments. The manuscript covers many literatures, mechanism of sex-dependent responses to checkpoint blockade. This review is well organized, but raises several questions.

  1. The review covers many hormones and immune pathways, but the mechanistic link between these and clinical PD-1/PD-L1 responses is not always clearly delineated.
  2. The relationship among sex, hormones, and the efficacy of immune checkpoint inhibitors is not clearly explained.
  3. In Section 2.8 and 2.9, mixed results and contradictory data were provided, but it is difficult to follow. It would be better organized and easier to understand if summarized in figures or tables.

Author Response

We thank the reviewer for the efforts to improve our manuscript. Please find a point-by-point response to the issues raised below.

Comment 1 - The review covers many hormones and immune pathways, but the mechanistic link between these and clinical PD-1/PD-L1 responses is not always clearly delineated.

Response 1 – There is not real concrete mechanistic link between the sex hormones and response to PD1/PDL1 therapies.  The connections are theoretical and speculative, and where there is no evidence, we have stated that in the manuscript. We have more clearly delineated these connections in section 2.13 and lines 658 to 676.

Comment 2 - The relationship among sex, hormones, and the efficacy of immune checkpoint inhibitors is not clearly explained.

Response 2 – There is no real understanding of what is driving this connection. It is only theoretical. We theorize on the causes for these connections in sections 2.9 and lines 315 to 324 to describe the sex differences in PD-L1 expression or the description of sex hormones in PDL1 efficiency in section 2.13 and lines 658 to 676.

Comment 3 - In Section 2.8 and 2.9, mixed results and contradictory data were provided, but it is difficult to follow. It would be better organized and easier to understand if summarized in figures or tables.

Response 3 – We have added Table 2.

Reviewer 2 Report

Comments and Suggestions for Authors

In this study the authors address an important topic about Sex-Differences in the Response to Lung Cancer and its Relation to PD-1/PDL-1 Checkpoint Therapies. Despite this some points should be addressed before publishing.

Title should be devoid of undefined acronyms.

Abstract, aim of the study and methods parts unclear.

Introduction, please add information about lung cancer risk factors, epidemiology and mortality rate. Moreover add details about the effect of sex on malignant transformation of lung cells in the term of smoking, alcohol drinks and sex hormones. As well add details about lung cancer therapeutic modilities and emphasized immunotherapy and it's advantages. Moreover add examples of immunotherapy. At the end of the introduction section please add the study rationale and aims in a clear statements.

Methods, please add methods section illustrated the type of study narrative, systematic review, etc. Moreover, add academic engines used in the data collection such as Pubmed Google scholar Scopus Medline etc, alongside software used in the manuscript preparation.

Results, please add table indicating the precise roles of sex hormones in immunity and cancer.

Discuss, please refine this section and support your work with similar finding and add the contradictory if present. Moreover discuss sev hormones signals in oncogenesis and immunity. Please discuss the study limitations. Please the study novelty.

Please check the manuscript for misuse of acronyms.

Please check the manuscript for long sentences or paragraphs without references citation.

Please check the references list for 2025  citation dated.

Comments on the Quality of English Language

Please check the manuscript for minor grammar errors and syntax.

Author Response

We thank the reviewer for the efforts to improve our manuscript. Please find a point-by-point response to the issues raised below.

Comment 1 - Title should be devoid of undefined acronyms.

Response 1 – The title has been corrected.

Comment 2 - Abstract, aim of the study and methods parts unclear.

Response 2 – We have rewritten the abstract.

Comment 3 - Introduction, please add information about lung cancer risk factors, epidemiology and mortality rate. Moreover add details about the effect of sex on malignant transformation of lung cells in the term of smoking, alcohol drinks and sex hormones. As well add details about lung cancer therapeutic modilities and emphasized immunotherapy and it's advantages. Moreover add examples of immunotherapy. At the end of the introduction section please add the study rationale and aims in a clear statements.

Response 3 – We have rewritten the introduction section to incorporate these comments. Please refer to introduction and lines  45 to 85.

Comment 4 - Methods, please add methods section illustrated the type of study narrative, systematic review, etc. Moreover, add academic engines used in the data collection such as Pubmed Google scholar Scopus Medline etc, alongside software used in the manuscript preparation.

Response 4 – We have updated the methods section to incorporate these requested changes. Please refer to lines 86 to 93.

Comment 5 - Results, please add table indicating the precise roles of sex hormones in immunity and cancer.

Response 5 – Please see Table 3.

Comment 6 - Discuss, please refine this section and support your work with similar finding and add the contradictory if present. Moreover discuss sev hormones signals in oncogenesis and immunity. Please discuss the study limitations. Please the study novelty.

Response 6 – We have rewritten the discussion section to incorporate these results. Please refer to lines 677 to 673.

Comment 7 - Please check the manuscript for misuse of acronyms.

Response 7 – We have corrected several incorrect acronyms including PD1 and PDL1.

Comment 8 - Please check the manuscript for long sentences or paragraphs without references citation.

Response 8 – We have corrected several of the long stretches of text with missing references.  These are located throughout the text.

Comment 9 - Please check the references list for 2025  citation dated.

Response 9 – We added some references published in 2025.

Round 2

Reviewer 1 Report

Comments and Suggestions for Authors

Revised manuscript has been well written based on reviewer's comment.